# Hematopoietic Cell Transplantation for Systemic Sclerosis—A Review

**DOI:** 10.3390/cells11233912

**Published:** 2022-12-03

**Authors:** Daniel Levin, Mohammed S. Osman, Caylib Durand, Hyein Kim, Iman Hemmati, Kareem Jamani, Jonathan G. Howlett, Kerri A. Johannson, Jason Weatherald, Matthew Woo, Jason Lee, Jan Storek

**Affiliations:** 1Cumming School of Medicine, University of Calgary, Calgary, AB T2N 4N1, Canada; 2Faculty of Medicine, University of Alberta, Edmonton, AB T6G 2R7, Canada; 3Faculty of Medicine, University of British Columbia, Vancouver, BC V6T 1Z3, Canada

**Keywords:** systemic sclerosis, hematopoietic stem cell transplant, sclerosis, ILD

## Abstract

Systemic sclerosis (SSc) is an autoimmune, multi-organ, connective tissue disease associated with significant morbidity and mortality. Conventional immunosuppressive therapies demonstrate limited efficacy. Autologous hematopoietic stem cell transplantation (HCT) is more efficacious but carries associated risks, including treatment-related mortality. Here, we review HCT as a treatment for SSc, its efficacy and toxicity in comparison to conventional therapies, and the proposed mechanisms of action. Furthermore, we discuss the importance of and recent developments in patient selection. Finally, we highlight the knowledge gaps and future work required to further improve patient outcomes.

## 1. Introduction

Systemic sclerosis (SSc) is a multi-organ, autoimmune, connective tissue disease. It is relatively rare, with global incidence reported as 8–56 new cases per million and prevalence as 38–341 cases per million [1]. Patients with SSc live an average of 20 years shorter than age- and sex-matched healthy controls. While SSc is eight–nine times more common in women, outcomes are worse in men [2,3].

The extent of skin tightness/fibrosis is probably the most important prognostic marker. About two-thirds of patients have limited cutaneous SSc (lcSSc), and one-third have diffuse cutaneous SSc (dcSSc), the latter having a poorer prognosis [4,5,6,7,8,9]. Patients with dSSc have both proximal and distal skin thickening and have tendon friction rubs, interstitial lung disease, heart disease, diffuse gastrointestinal disease (both of the esophagus and the more distal digestive tract), and Scl70 autoantibodies, more frequently compared to patients with lcSSc. Patients with lcSSc have only distal skin thickening and have severe Raynaud’s phenomenon, pulmonary hypertension, esophageal disease without distal digestive tract involvement, and anticentromere antibodies more frequently than dcSSc patients. As lcSSc has a better prognosis with conventional therapy than dcSSc, and hematopoietic cell transplantation (HCT) is risky, it is controversial whether lcSSc patients should be treated with HCT. Thus, we focus on dcSSc here.

SSc is associated with significant morbidity and mortality [10]. The most recent meta-analysis, which did not include HCT recipients, reports high standardized mortality ratios for dcSSc patients (3.7–6.1) and a mean standardized mortality ratio of 4.7 [11]. This reflects both the severity of SSc and the low efficacy of conventional (non-HCT) immunomodulatory therapies [12,13,14]. Delayed diagnosis and the progression of cardiopulmonary involvement are major drivers of mortality [11,15].

HCT consists of conditioning, graft infusion, and subsequent supportive care [16]. Conditioning contains immunosuppressive chemotherapy, with or without total body irradiation and antilymphocyte antibodies. The rationale for giving antilymphocyte antibodies is to enhance the killing of autoreactive T or B cells and, in the allogeneic HCT setting, to prevent graft-versus-host disease (GVHD). The graft can be either allogeneic (from a related donor or an HLA-matched unrelated donor) or autologous, using a patient’s own hematopoietic cells collected weeks before the HCT. The supportive care aims to mitigate the toxicities (e.g., transfusions for anemia and thrombocytopenia or antibiotics for leukopenia-induced infections) (Figure 1).

HCT has been explored in the past 30 years as a potential treatment for autoimmune diseases including SSc, initially with allogenic HCT (alloHCT)[17]. This was based on the proposition that alloHCT would replace an autoreactive immune system with a healthy immune system. It was also based on observations in rare patients with an autoimmune disease and a concurrent hematologic disease necessitating alloHCT, showing that the autoimmune disease usually remitted [18]. However, it was subsequently noted that in patients with an autoimmune disease and a concurrent hematologic malignancy necessitating autologous HCT (autoHCT), remissions of autoimmune diseases occurred even after the autoHCT. Concerns surrounding alloHCT toxicity, in particular GVHD [19,20], subsequently favored autoHCT as a less toxic alternative. In the last three decades, autoHCT for autoimmune disease has been reported in case reports, case series, prospective pilot/phase 1/phase 2 trials, and ultimately prospective phase 3 randomized trials. For SSc specifically, multiple non-randomized and three randomized trials [21,22,23] all showed superiority over conventional immunosuppressive therapy, establishing autoHCT as an effective therapeutic intervention in SSc. Additionally, three systematic reviews [24,25,26] support that HCT provides a survival benefit and results in a faster improvement of skin tightness and a mild improvement of lung function (see Clinical Results section for details). AutoHCT will be the focus of this review (hereafter, “autoHCT” and “HCT” will be used synonymously); however, a small section on alloHCT is also included.

Despite the high-level evidence demonstrating benefit in severe, rapidly progressing SSc [13], HCT use for SSc remains relatively infrequent [27], likely related to high HCT-related mortality (6–10%) [28], unfamiliarity with the treatment modality, and limited resources. There have been over 600 HCTs for SSc reported to the European Society for Blood and Marrow Transplantation (EBMT) registry [29] and over 100 to the North America-based Center for International Blood and Marrow Transplant Research (CIBMTR) registry [30]. Unfortunately, due to underreporting to the registries, a lack of information on HCTs performed outside of Europe and North America, and an unknown total denominator of global SSc cases (even for Europe and North America), the frequency with which HCT is offered to SSc patients cannot be assessed.

The goal of this review is to summarize the current knowledge on HCT for SSc including its proposed mechanisms of action, while highlighting knowledge gaps with this treatment modality and discussing future areas of research in the field.

## 2. Proposed Mechanisms of Action of Autohct on SSc

Immune reconstitution after autoHCT for SSc follows a similar pattern as after HCT for malignant diseases (the most frequent indication for HCT) [29,31,32,33]. Innate immune cell (neutrophil, monocyte, dendritic cell, NK cell) counts recover within weeks after HCT. Counts of B cells and CD8 T cells recover over months, whereas CD4 T cells over years after HCT. Most T cells in the first several months after HCT are memory/effector-type, derived from T cells that survived conditioning chemo/radiotherapy. Naive T cells start being produced by the thymus de novo (from stem cells) only later, primarily in younger patients. Therefore, T cell repertoire (clonal diversity) is restricted early post-transplant and diversifies only months to years later. The diversification is prominent in younger patients but is limited in older patients. Regarding B cells, naive B cells recover first but memory B cells only recover later. Nevertheless, production of antibodies against the microorganisms encountered pre-transplant and autoantibodies is not interrupted after HCT, as these antibodies are produced by long-lived, chemo/radiotherapy-resistant plasma cells. Regarding SSc-related autoantibodies, studies have shown a mild and inconsistent decline of Scl70 levels over years post-transplant, [31,34,35,36] a finding probably not associated with clinical response [34].

Despite much having been learned about the immune reconstitution after HCT, why autoHCT results in the remission of SSc remains unknown. This is due in part to the fact that the immunopathogenesis of SSc is poorly understood, so we do not know which crucial immune abnormalities (if any) need to be corrected. The evidence that SSc is an autoimmune disease is only circumstantial, derived from the fact that cellular infiltrates in the skin of patients with early scleroderma contain T cells, which are oligoclonal, B cells, and macrophages [37,38,39,40]. Nevertheless, multiple mechanisms of action of HCT have been proposed (summarized below).

Complete lymphoablation is followed by T cell generation de novo (thymopoiesis), assuming that autoreactive T cell clones generated by the thymus pre-transplant are not generated post-transplant. 

This was hypothesized in the 1990s, but it turned out not to be plausible, as it is impossible to kill all T cell clones even when using highly lymphocyte-toxic conditioning (TBI + cyclophosphamide + ATG) and a T cell depleted graft [41]. Moreover, HCT is clinically effective even when using low-intensity conditioning and no graft T cell depletion [28].

### 2.1. Partial Lymphoablation Followed by the Crowding out of the Persisting Autoreactive T Cells by De Novo Generated T Cells

This is difficult to prove, as autoreactive T cells causing SSc are putative. Nevertheless, it is known that T cell clones present in patients before HCT persists after HCT [41,42,43], and the clones that were quantitatively dominant before HCT become subdominant after HCT, possibly due to being crowded out by de novo generated T cells [43]. Thus, robust thymopoiesis might result in the subdominance of pre-transplant dominant autoreactive clones. Consistent with that, both in SSc and multiple sclerosis, a high likelihood of relapse/progression-free survival or a low likelihood of relapse/progression appears associated with a younger age [44,45,46,47] and with increased T cell clonal diversity after HCT [48,49,50]. However, the data on T cell clonal diversity is based on small numbers of patients (in particular, small numbers of relapsing/progressing patients) and, in the case of multiple sclerosis, on only an early time point (2 months post-HCT) and has not been universally reproduced [51], so further studies are needed to evaluate this hypothesis.

### 2.2. Partial Lymphoablation Followed by the Functional Tolerization of Autoreactive T Cells

Though not evaluated specifically for SSc, after HCT for other diseases, the reconstituting T cells appear to underproduce proinflammatory cytokines, such as interferon-gamma (IFNg) or tumor necrosis factor-alpha (TNFa), when stimulated in vitro by a polyclonal mitogen [52,53]. Consistent with that, the IFNg-related transcripts in the blood from SCOT trial patients were increased (compared to healthy controls) at baseline and decreased to near healthy control levels in the patients who received HCT, whereas they remained increased in the patients who received conventional (minimally effective or ineffective) therapy [54]. The serum levels of TNFa were also found to be high in SSc patients pre-HCT and significantly declined over 1 to 3 years post-HCT [35]. Furthermore, expression of PD1 (checkpoint inhibitor of T cell activation) on the T cells of SSc patients was found to be increased after HCT (compared to pre-HCT) in responders but not in non-responders [47]. The functional tolerization of T cells could be facilitated by regulatory T cells (Tregs) or myeloid cells (reviewed next).

### 2.3. Fast Reconstitution of Tregs

The faster reconstitution of regulatory T cells than conventional T cells after HCT has been documented in multiple settings and appears to promote tolerance, as extensively reviewed by Hendrawan et al. [55]. In SSc patients, the reconstitution of Treg (FoxP3^+^ CD25^high^ CD4^+^ T cell) counts was faster in HCT responders vs. non-responders [47]. Not only the quantity but also the function of Tregs from SSc patients may be better after than before HCT, as assessed by the inhibition of the proliferation of polyclonally (anti-CD3 and anti-CD28) stimulated conventional T cells [56]. However, despite autologous-like HCT improving clinical manifestations of proteoglycan-induced arthritis in mice, the addition of FoxP3^+^ Tregs to the graft did not result in further clinical improvement [52]. Whether further clinical improvement of an autoimmune disease could be achieved by infusions of Tregs weeks or months after HCT should be evaluated in animals.

### 2.4. Less Proinflammatory/More Anti-Inflammatory Myeloid Cells? 

Monocytes from patients at ~3 months after HCT for various diseases, but not from healthy controls, were shown to suppress polyclonal (anti-CD3-stimulated) T cell proliferation [57]. Neutrophils may also play a role in the mechanism of action of HCT for autoimmune diseases. In the blood of SCOT study patients, neutrophil transcripts were increased (compared to healthy controls) at baseline and normalized after HCT but not after conventional therapy [54].

### 2.5. Role of B Cell?

The role of B cells in the HCT mechanism is uncertain. In one study in SSc patients, quantitative B cell reconstitution was faster in responders than non-responders [51], but this requires validation. Regulatory B cell (Breg) counts were evaluated in two studies of SSc patients undergoing HCT [48,58]. Both found higher Breg (CD24^high^ CD38^high^ B cell) counts at 1 year in responders vs. non-responders. Bregs may also be more functional after than before HCT. Findings that suggest this include changes to in vitro IL-10 production with CpG-stimulated purified patient total B cells [59] and by inhibition of IFNg and TNFa production with in vitro stimulated CD4 T cells [58]. However, if Bregs play a significant role, it is difficult to explain why rituximab appears effective in patients who relapsed after HCT [60,61], with rituximab theoretically expected to eliminate the few Bregs that the relapsing patients may still possess. One such explanation could be that rituximab does not in practice deplete all B cell populations, as has been seen with marginal zone B cells [62].

### 2.6. Non-Immunologic Mechanisms?

Our group showed that DNA mutations similar to those in cancer cells are present in the skin (probably fibroblasts) of SSc patients and that these mutations likely translate into immunogenic neoantigens [63]. Thus, it is conceivable that SSc is a cancer-like disease and that the immune response to the neoantigens, suggested by the presence of oligoclonal T cells in skin, is secondary [39]. Therefore, whether the primary mechanism of HCT is the killing of the cancer-like skin cells (fibroblasts?) by the conditioning chemo/radiotherapy should be evaluated.

## 3. Clinical Results

Conventional immunosuppressive therapies such as cyclophosphamide (CYC), mycophenolate mofetil (MMF), and methotrexate (MTX) are currently recommended as the first-line therapies for SSc, despite limited efficacy [64]. In Appendix A, we summarize the results of randomized studies of conventional and newer non-HCT therapies. In addition, we have included the European Scleroderma Observational Study (ESOS), which is a large, retrospective, registry-based comparison of CYC, MMF, MTX, and no immunosuppression [65]. With the possible exception of nintedanib and rituximab, for which long-term follow up data are unavailable, the results of the studies in Appendix A highlight the limited efficacy of non-HCT therapy for improving morbidity and mortality. It is also interesting that even recent studies have used placebos for controls, further highlighting the limited efficacy of conventional therapies.

In the last decade, three randomized trials comparing conventional therapy of SSc (cyclophosphamide) to HCT (ASSIST, ASTIS, SCOT) have been published [21,22,23]. Despite differences between patient populations and HCT techniques, all three trials demonstrated the superiority of HCT over conventional immunomodulation with cyclophosphamide. These randomized trials and one case-control retrospective study are summarized in Appendix A. The case-control study, while retrospective and not randomized, is included as it directly compared the disease outcomes of 18 patients undergoing HCT to 36 matched controls treated conventionally [66]. The results of the case-control study and the three randomized studies can be summarized as (1) skin tightness, as measured by the modified Rodnan skin score (mRSS) improving in HCT recipients vs. worsening or improving to a lesser degree in controls (Figure 2) [67], (2) lung function mildly improving in HCT recipients vs. mildly worsening in controls (Figure 3) [68], and (3) quality of life (QOL) improving in HCT recipients vs. worsening or improving to a lesser degree in controls (Figure 4). The ASTIS and SCOT trials also evaluated the impact on overall survival (OS) and event-free survival (EFS, i.e., survival free of organ failure). In both studies, the OS and EFS at ≥4 years were higher in HCT recipients vs. controls (Figure 5) [22,23].

Transplant-related mortality at 1 year was 0% in ASSIST, 10% in ASTIS, and 3% in SCOT (6% if a late nonrelapse death is counted). It was >10% in the 1990–2010 era, when 8/34 (23%) of patients in a Seattle-led multi-institutional phase 2 study died by 1 year [69]. More recently, this number has fallen to <10%, e.g., 6.3% among 80 patients in a prospective non-interventional multi-institutional European study [44], 5.6% among 18 patients in a single-center study from Italy [66], and 4.2% among 24 patients (surviving patients counted only if ≥1 year follow-up) at our center (unpublished). This is likely attributable in part to improved patient selection (e.g., excluding patients with advanced cardiopulmonary disease) and to improved techniques (e.g., shielding lungs and kidneys when total body irradiation is used in conditioning), and increased center-specific experience [70].

The effects of HCT on multiple manifestations/aspects of SSc and specific organs are summarized in Table 1 and covered in detail in the Appendix A online.

## 4. Patient Selection

Appropriate patient selection for HCT is critical. Cardiac disease is known to adversely affect HCT safety, as the combination of high-dose cyclophosphamide (with its known cardiotoxicity), chemotherapy-associated intravenous hydration (with its associated volume overload), and anti-thymocyte globulin-associated vasodilation [106] place excessive strain on a cardiopulmonary system with low reserve [107]. In early stages of SSc, cardiac impairment is often underappreciated by echocardiography [108]. Mortality has been largely attributed to cardio/pulmonary events, with recognition that baseline ECG, echocardiogram, and resting right heart catheterization are insufficient to assess complete cardiac risk. In the last decade, the addition of cardiac magnetic resonance imaging and right heart catheterization with fluid challenge or exercise have become mainstays in screening and appear associated with improvements in transplant-related mortality in recent studies [107]. Based on careful clinical observations and clinical judgment opinions, inclusion and exclusion criteria have been created for both trials and standard clinical practice. These have varied among studies and vary among centers, though typically include the criteria shown in Table 2.

Despite the conventional selection criteria, recent reports of successful HCT transplants in SSc patients who would have been excluded from randomized studies primarily due to poor cardiopulmonary status [28,109] highlight the heterogeneity of patient response and the need to refine inclusion criteria. In particular, when the combination of fludarabine, low-dose cyclophosphamide, ATG, and rituximab is used instead of high-dose cyclophosphamide-based conditioning [28], more liberal criteria can likely be applied (Table 2). With this conditioning, the transplant-related mortality in high-risk patients was only 2.4%, which is lower than what was seen in the randomized control trials (ASTIS, ASSIST, SCOT). It is presumed to be secondary to lower doses of cardiotoxic cyclophosphamide. Additionally, a shorter period of neutropenia (5 days) may have contributed to the low transplant-related mortality with this regimen.

The demographic and clinical risk factors for disease progression or poor progression-free survival after HCT include male sex, older age, smoking history, high mRSS score (>24 or >25) and LVEF <50% [22,23,44,45,75]. It is unclear whether these variables should guide patient selection for HCT, as the same risk factors may apply for SSc patients not treated with HCT [15,110].

Our understanding of molecular risk factors remains limited. To characterize which patients are most likely to benefit from HCT, one study employed machine learning to analyze subsets of SSc patients involved in the SCOT trial, based on their leukocyte transcriptome [111]. Patients were divided into fibroproliferative, inflammatory, and normal-like transcriptome subsets. EFS was superior after HCT vs. conventional therapy for patients with fibroproliferative disease, trended to be superior for patients with inflammatory disease, and was similar after HCT and conventional therapy for patients with normal-like disease. Alternatively, human leukocyte antigens (HLA) were explored in 46 HCT recipients [112]. Most responders appeared to have HLA-E*01:03 or HLA-G 14bp del alleles, whereas most non-responders carried a double dose of the HLA-B threonine leader peptide. While preliminary, these studies suggest that nucleic acid testing of patients prior to HCT may be a valuable aspect of patient selection in determining the benefits vs. the risks of the procedure.

The HCT comorbidity index (HCT-CI), a weight score of 15 pre-transplant comorbidities, predicts transplant-related mortality in patients with hematologic malignancies. However, this does not appear to be the case for SSc [45].

SSc overlapping with another autoimmune disease poses a problem for patient selection, as the randomized studies generally accrued only patients with pure SSc. Moreover, it is difficult to determine whether a condition such as myositis, arthritis (synovitis), or Sjogren’s syndrome is a manifestation of SSc or represents an overlap with another disease. Presence of anti-citrullinated peptide or rheumatoid factor antibodies suggests an overlap with rheumatoid arthritis (RA), and the presence of anti-double stranded DNA or anti-Smith antibodies suggests an overlap with systemic lupus erythematosus (SLE) [113]. At our center, two female patients underwent HCT who had SSc overlapping with SLE. One of them experienced relapse of both SLE and SSc (including worsening skin tightness/contractures) at 6 months post-transplant, whereas the other one experienced relapse of SLE, which was mild, but not SSc (as her skin tightness/contractures improved) as of 3 years post-transplant.

## 5. Transplant Protocols

AutoHCT consists of hematopoietic cell mobilization from marrow to blood, collection of the hematopoietic cells from blood by mononuclear cell apheresis, potential graft manipulation (e.g., depleting T and B cells by enriching for CD34+ cells), graft cryopreservation, conditioning (chemo/radiotherapy, typically with anti-lymphocyte antibodies), infusion of the graft, and post-transplant supportive care to prevent or treat acute complications of the chemo/radiotherapy (e.g., antimicrobial drugs or transfusions)[114]. Different methods of mobilization, graft manipulation, conditioning, and supportive care have been employed. These are reviewed below. Unfortunately, a comparison of the different conditioning regimens (possibly the most important component of HCT) used in the prospective randomized trials cannot be conducted, due to differences in patient populations and in treatment factors other than the conditioning (Appendix A). No method of mobilization, graft manipulation, conditioning, or supportive care has been conclusively shown to be superior to another method, with no head-to-head comparisons.

### 5.1. Mobilization

As granulocyte-colony stimulating factor (G-CSF) may exacerbate some autoimmune diseases, cyclophosphamide (CYC) is typically used in conjunction with G-CSF for mobilization. Interestingly, G-CSF-only mobilization was used in SCOT [23], and no exacerbations of SSc were noted. CYC was used at 4 g/m^2^ in ASTIS [22]; however, the dose was recently challenged by the NISSC1 study [44], which demonstrated an adequate CD34 cell yield with only 2 g/m^2^ of CYC (as was used in ASSIST). This is consistent with a retrospective study by Pecher et al., which showed no difference in stem cell yield in a CYC dose of 4 g/m^2^ vs. 2 to 3 g/m^2^ [115]. Theoretically, the use of the lower CYC dose could limit cardiotoxicity.

### 5.2. Graft Manipulation/CD34 Cell Selection

The rationale for CD34 selection is to eliminate most lymphocytes, thus reducing the likelihood of auto-reactive lymphocytes being re-infused into the patient. CD34 selection has been used in many non-randomized trials and was used in two of the three randomized control trials (ASTIS, SCOT) [22,23]. However, the necessity for CD34 selection has more recently been questioned. In a retrospective EBMT analysis, no demonstrated benefit of CD34 selection was found in 138 SSc patients with similar baseline characteristics, with potential complications including increased infection rates [116]. Similarly, in a North American retrospective analysis, no significant differences in outcomes were found between patients whose grafts were CD34-selected or not [30]. Conversely, a European prospective non-interventional study and a Japanese retrospective study demonstrated superior treatment responses in patients who received CD34 selection, though the superior treatment responses did not translate into a significantly higher OS or PFS [44,117]. Thus, the role of CD34 selection or another graft manipulation requires further research.

### 5.3. Conditioning

*Cyclophosphamide/other chemotherapy.* CYC has been the backbone of the conditioning regimens used to date. Some studies have evaluated reduced-dose CYC with thiotepa or fludarabine, reporting similar treatment responses, including reductions in mRSS and improvements in lung function [25,89]. Burt et al. [28] used a “cardiac-safe regimen” with lower doses of CYC (60 mg/kg) in SSc patients with compromised cardiopulmonary function and found reduced transplant-related mortality than was seen in the three randomized studies (ASSIST, ASTIS, SCOT), likely due to the reduced cardiotoxicity of the conditioning and duration of neutropenia. A subsequent case report of two patients with SSc undergoing HCT, where 100–150 mg/kg CYC was successfully used to treat SSc with a long-term clinical response, further supports the idea that CYC doses may be lowered moving forward [100]. By doing so, there is also a potential reduction in the risk of CYC-associated cardiotoxicity. However, neither prospective nor retrospective studies have been conducted to compare the toxicity and efficacy of high-dose CYC (200 mg/kg) alone to lower-dose CYC alone or with another chemotherapeutic agent.

*Total body irradiation (TBI).* While TBI would be expected to increase transplant-related mortality given its immunosuppressive effect and its potential to induce myeloid as well as solid malignancies, the transplant-related mortality in the SCOT trial (using 8 Gy TBI with lung and kidney shielding to 2 Gy plus CYC 120 mg/kg) was relatively low (6% at 6 years) [23]. At present, there is no study guiding whether TBI-based conditioning is better, worse, or similar to chemotherapy-only conditioning. If TBI-based conditioning is used, it is likely important to shield the lungs and possibly the kidneys to minimize mortality due to pulmonary or renal failure [69].

*Anti-lymphocyte antibodies.* Antithymocyte globulin (ATG) most frequently, including in the three randomized studies (Table 2), while alemtuzumab has been used in only a minority of cases. The rationale for anti-lymphocyte antibodies is to maximize the depletion of alloreactive T or B cells [118]. It is not known whether this theoretical benefit is outweighed by the risk of side effects such as infusion reactions and infections.

ATG itself has been studied as a stand-alone DMARD for the treatment of SSc in three separate observational studies [119,120,121]. While limited efficacy was seen in improving the skin score for some patients, others worsened in both their skin scores and pulmonary function, in addition to suffering adverse events such as serum sickness. As such, the use of ATG as a stand-alone treatment for SSc is not currently recommended. By inference, the role of ATG in conditioning can be questioned. Two cases of HCT for SSc using ATG-free conditioning were reported [122]. Both patients had a clinical improvement in their SSc symptoms; however, the follow-up was short (2 and 4 months). It needs to be further studied whether ATG is needed.

Alemtuzumab (anti-CD52 antibody) is theoretically attractive as it depletes both T and B cells; thus, theoretically, it also autoreactives T and B cells. However, in one retrospective study, alemtuzumab was associated with increased mortality after HCT for SSc [123]. Additionally, alemtuzumab is associated with an increased risk of developing new autoimmune diseases after HCT [124]. As such, the use of alemtuzumab in conditioning is currently not recommended.

### 5.4. Supportive Care

Studies differed in peri-/post-transplant supportive care or did not report the details of supportive care (Appendix A). Therefore, the optimal supportive care of HCT for SSc is unknown. Generally, the same supportive care that is used with autoHCT for malignancies has been used, which varies among centers.

From the perspective of HCT for SSc, specifically, it should be highlighted that careful fluid management peri-transplant is prudent. Volume overload can lead to cardio/pulmonary failure, and dehydration could lead to renal failure. On the other hand, it is generally accepted that some intravenous fluid should be given with high-dose cyclophosphamide to minimize the risk of hemorrhagic cystitis [125]. Perhaps Mesna, which has been shown to be equally effective as intravenous fluids for the prevention of hemorrhagic cystitis [126], would provide similar protection without the risk of fluid overload.

## 6. Complications of HCT

### 6.1. Acute Complications

Acute complications of HCT include pancytopenia managed with transfusions and antimicrobials; nausea/vomiting managed with antiemetics; damage to the gastrointestinal tract epithelium leading to stomatitis, esophagitis, and enteritis (presenting with diarrhea); alopecia; and organ failure (heart, lungs, kidneys, liver (rarely)) [127]. The acute complications are typically manageable with supportive care and largely reversible, except for organ failure. This is an impetus for transplant physicians to select patients without severe heart, lung, kidney, or liver disease and for rheumatologists to refer patients for HCT before they have developed severe cardiac, pulmonary, or renal disease.

Despite adequate pre-screening, acute and fatal cardiotoxicity still occurs [128]. Both ASSIST and ASTIS used high-dose cyclophosphamide-based regimens (200 mg/kg) and found that acute cardiac failure during transplant hospitalization was a significant driver of transplant-related complications. While cyclophosphamide may play a role in this finding, underlying SSc-related cardiac involvement and significant fluid challenge during conditioning and early post-transplant infections may also play a role. Conditioning regimens with a lower dose of cyclophosphamide have been and need to be further evaluated.

Engraftment syndrome is thought to be caused by the autoinflammatory reactions of the regenerating leukocytes. First coined by Lee et al. [129], it is associated with fever, rash, and occasionally the pulmonary infiltration occurring in the first 2 weeks post autoHCT [130]. The risk factors for engraftment syndrome in SSc patients have been identified, including the use of G-CSF, SSc-associated cardiac involvement, and older age [131]. In one study of SSc patients post autoHCT, 41% developed engraftment syndrome, which in most cases was self-limited. For patients with severe or prolonged engraftment syndrome, a short course of prednisone is recommended based on experience in patients undergoing autoHCT for malignancies.

The immune deficiency associated with infections persists even after neutrophil recovery and usually recovers within 1 to 2 years. This is primarily due to the slow recovery of CD4 T cells, particularly in older patients. Prophylaxis of Pneumocystis jirovecii and Streptococcus pneumoniae infections with cotrimoxazole and zoster with valacyclovir is recommended until 1 to 2 years.

### 6.2. Long-Term Complications

These include infertility, new autoimmune diseases, and malignancies [127]. Here, we summarize for each complication what is known about its incidence and management in patients who underwent autoHCT for any disease. Where available, the data specific to SSc patients are included.

#### 6.2.1. Infertility/Premature Menopause

Infertility is a common side effect of HCT for cancer. In a study of 639 HCT recipients (39% autologous, 53% males, age 21–45 years, surviving ≥2 years after HCT), conception was 36 times less likely compared to their siblings [132]. Risk factors for non-conception were older age (≥30 years), female sex, and TBI.

However, after HCT for autoimmune diseases, the incidence of premature menopause or infertility appears lower than in the oncology setting. Menses resumed at a median of 7 months in 70% premenopausal women with multiple sclerosis [133] and at a median of 4 months in 35% premenopausal women with various autoimmune diseases [134]. In both studies, the likelihood of resuming menses appeared higher in younger ( < 30 y old) than older women. In the first study [133], four women with resumed menses tried to conceive—three successfully (age 31, 32, 33) and one unsuccessfully (age 38). In an EBMT study of 324 women (age 18–50 y) who underwent HCT for various autoimmune diseases, 20 natural conception pregnancies were reported, but only 13 resulted in live births (at median age 32 years), 6 of them by Caesarian section. The seven failed pregnancies were due to spontaneous abortion (*n* = 4), ectopic pregnancy (*n* = 1), spontaneous fetal death (*n* = 1), and induced abortion (*n* = 1). An exacerbation/relapse of autoimmune disease occurred in two patients during their second pregnancies. There were no reports of congenital, developmental, or any other disease in the children. No maternal mortality was associated with pregnancy or postpartum. Six of the natural conception pregnancies occurred in five women with SSc. Of the six pregnancies, one resulted in natural delivery, four in Caesarian section delivery, and one in spontaneous abortion. In a single-center study from Berlin, Germany, of 15 women (age 22–48 y) with various autoimmune diseases, of whom 4 delivered children after HCT (1 of them twice), 2 had pregnancies that were complicated with preeclampsia, 1 with premature labor (at 31 weeks), 1 with breech presentation, and 1 with no complications [135]. Among the 15 women, there was only 1 woman with SSc who was not amenorrheic pre-HCT. She was 22 years old at HCT and had two deliveries after HCT (at ages 23 and 24 years). Both were complicated (one premature labor and one breech presentation). Taken together, these findings suggest that female infertility/premature menopause after HCT for autoimmune diseases may not be as frequent as after HCT for cancer; however, this has not been studied specifically for patients with SSc. Complications of pregnancies and deliveries are frequent.

Male fertility after HCT for autoimmune diseases has not been adequately studied. A Charite University (Berlin) study [135] included four men, of whom three had normal levels of follicle-stimulating hormone (FSH) and testosterone after HCT, whereas one had levels consistent with infertility. Only one of them had SSc, and he was among the three men with normal post-transplant FSH and testosterone levels. Studies are needed on the live sperm counts and the percentage of males whose female partners became pregnant among couples that wished to conceive.

#### 6.2.2. Malignancy

New malignancies are a well-defined complication of HCT for hematologic cancer [136]. The reported cumulative incidence of solid tumors has ranged from 1–2% at 5 years to 4–15% at 15 years post-transplant, without a plateau (the incidence is steadily increasing). The incidence is about twice as high as in the general population. Likewise, the incidence of myeloid malignancies (e.g., myelodysplastic syndrome or acute myeloid leukemia) after autoHCT has been steadily increasing over time after HCT, but the relative incidence is much higher than for solid tumors (10- to 20-fold higher than in the general population). The absolute incidence is about 3% at 5 years and 6% at 10 years [137]. The main risk factors for both solid tumors and myeloid malignancies are radiation and chemotherapy, both before and during conditioning.

Little is known about new malignancies after HCT for SSc. The evaluation is complicated by the fact that, compared to the general population, SSc patients have an increased baseline incidence of solid tumors (breast, lung) or hematologic malignancies, even without HCT [138,139]. Cyclophosphamide is known to cause cancer, particularly bladder and blood cancer [140,141], and is used not only in HCT (for stem cell mobilization and transplant conditioning) but also as the conventional therapy for SSc. Theoretically, the malignancy risk after HCT for SSc should be lower than after HCT for hematologic malignancy, as SSc patients typically do not get intensive chemo/radiotherapy pre-transplant (before conditioning). Consistent with this, no malignancies were reported after HCT vs. three reported malignancies after conventional cyclophosphamide in the ASTIS study. In the SCOT study, three malignancies were reported after HCT (2 myelodysplastic syndromes, 1 thyroid cancer) vs. one (breast cancer) after conventional cyclophosphamide. Long-term follow-up studies are needed to determine if HCT is associated with an increased risk of cancer in patients with SSc.

#### 6.2.3. New Autoimmune Disease 

New autoimmune diseases occur in about 10% of HCT recipients for autoimmune diseases, typically months after HCT [142]. These are called secondary autoimmune diseases (sAD), although causality has not been established. Genetic predisposition to auto-immunity, T cell reconstitution allowing self-reactive T cells to expand (due to lymphopenia-induced homeostatic proliferation), and accumulation of mutations in lymphocytes during post-HCT proliferation have been proposed as mechanisms [124]. An EBMT study found a 5-year cumulative incidence of sAD, among 347 patients who underwent HCT for their primary autoimmune disease, to be 9.8% [142]. Observed sADs included autoimmune hemolytic anemia, acquired hemophilia, autoimmune thrombocytopenia, antiphospholipid syndrome, thyroiditis, Graves’ disease, myasthenia gravis, rheumatoid arthritis, sarcoidosis, vasculitis, and psoriasis. Risk factors for sAD after HCT have been determined in only two studies: Daikeler et al. identified three risk factors—SLE as the primary disease, a short interval from AD diagnosis to HCT, and graft enrichment for CD34 cells in combination with ATG [142]. Loh et al. identified one risk factor: the use of of alemtuzumab in conditioning (as opposed to ATG or no lymphodepleting antibody) [124].

In SSc specifically, post-HCT sAD incidence has been reported at 9% [143] and 27% [131], with median onset at 1–2 years after HCT. A possible explanation for why a high (27%) sAD incidence was seen in the study of Strunz et al. was their use of CD34 depletion together with a high dose of ATG (30 mg/kg). Interestingly, in the Strunz study, ATG-Grafalon was used as opposed to Thymoglobulin (used in most other studies) (Joerg Henes, personal communication, 11 September 2022).

## 7. Relapse/SSc Progression

### 7.1. Definition

The definition of what constitutes SSc relapse after HCT remains controversial. Most studies considering disease progression or relapse, which are often used interchangeably, have included worsening mRSS (e.g., increased by >25% and >5 points) or FVC (e.g., ≥10% relative decline in FVC)[45] or the need for immunomodulatory drug initiation for relapse/progression. The European prospective non-interventional study [44] used a more elaborate definition, taking into account also a significant drop in LVEF, body weight, creatinine clearance, or quality of life. A standardization of relapse definition is needed. A standardized definition, e.g., as proposed by the OMERACT initiative [144], will be important to include in future *studies.*

### 7.2. Frequency of Relapse

Better disease control and EFS when compared to controls were seen in all three randomized trials (ASSIST, ASTIS, SCOT). For example, in the SCOT trial, 9% of HCT recipients vs. 44% of controls started an immunomodulatory drug within the first 4.5 years [23], with the difference remaining significant at 6–11 years [95]. Aside from the randomized trials, disease progression varies among longitudinal studies but has been reported to occur in 11% to almost 31% [44,45,60,145] of patients at 4.4 years, 10.5% to 24% at 8 years [107,117], and 42% at 11 years [75] after HCT. While the reasons for this heterogeneity are unclear, it is likely due to differences in patient selection, follow-up duration, and the definition of relapse used. Unfortunately, no plateau on the incidence of relapse/progression curve may be achieved over 3 to 5 years post-transplant [44,70]. Further work into determining which patients are more likely to relapse is required to make better informed decisions surrounding patient selection.

### 7.3. Treatment of Relapse

Currently, there is no evidence to guide the optimal therapy of SSc relapse after HCT. Patients are usually treated with conventional immunomodulatory drugs, but their efficacy in the post-transplant setting has not been formally evaluated. In RA patients who had relapsed after autoHCT, rituximab resulted in clinical improvements similar to their initial responses to autoHCT in 80% of patients [61]. Subsequently, a single-center analysis of nine HCT recipients who received rituximab for SSc relapse after HCT (n = 6) or for a new autoimmune disease (n = 3) showed improved mRSS (compared to no improvement in seven patients who received no immunomodulatory therapy after relapse) and stabilization of lung function [60]. Alternatively, a second autoHCT was successful for treating SSc relapse in one reported case [146].

## 8. Allogeneic HCT

While alloHCT is more toxic than autoHCT (due to GVHD or graft rejection), there are theoretical arguments for its use, including the need to more completely replace an auto-reactive immune system with a healthy one. For autoimmune diseases in general, survival after alloHCT significantly improved between 2000–2004 and 2011–2015 [70]. Moreover, new HCT protocols with low transplant-related mortality have been developed in the 2010s for hemoglobinopathies and aplastic anemia [147,148,149,150,151]. This has renewed interest in alloHCT for autoimmune diseases. Nevertheless, at present, alloHCT for autoimmune diseases remains largely restricted to pediatric practice due to the higher efficacy [152] and theoretically lower transplant-related mortality in children than adults.

For SSc, four cases of alloHCT have been published [19,153,154]. Remission was achieved in all four cases and lasted until the end of follow-up (median 2.7 years after alloHCT and median 1.2 years after discontinuation of immunosuppressive therapy for GVHD prophylaxis/treatment). Surprisingly, complete chimerism (all hematolymphatic cells of donor origin) was not a prerequisite for remission, as has also been noted for other autoimmune diseases [155].

## 9. Future Works

### 9.1. HCT as First vs. Next Line Treatment?

While HCT has been well-established as a superior treatment modality for rapidly progressing SSc, whether HCT should be an upfront or rescue treatment for patients not responding to conventional therapies remains unclear. We eagerly await the results of the currently ongoing, randomized, open-label trial (DIFFUSE) designed to answer this question [156].

### 9.2. Refining Patient Selection

This could be based on demographic/clinical or laboratory characteristics. At present, no clinical or demographic characteristics are conclusively shown to be helpful. Despite the typical exclusion criteria for HCT, such as advanced age or poor vital organ function, the patients ineligible for HCT based on these criteria also have very poor outcomes with conventional therapy [111]. It is unknown whether in these patients the high risk of transplant related mortality is outweighed by the anti-SSc effect of HCT, particularly when a low-toxicity conditioning is used [28]. It will be important to retrospectively compare the outcomes of the subgroups of patients with various demographic/clinical characteristics receiving HCT vs. conventional therapy, using registries or large institutional databases. As HCT strategies continue to be refined in the future, the possibility of including such patients with a higher comorbid risk, such as mild diastolic dysfunction, for instance, may prove more beneficial with fewer risks. Regarding laboratory selection, we praise the work of Franks et al. [111], which shows that patients with a fibroproliferative and possibly an inflammatory transcriptome are more likely to benefit from HCT than conventional therapy, whereas patients with a normal-like transcriptome are likely to benefit equally from either treatment. It will be important to validate this biomarker and search for additional biomarkers that could help refine patient selection.

HCT for lcSSc will also need to be explored if transplant-related mortality continues to decline.

### 9.3. Comparison of VARIABLES in HCT Techniques and Development of New HCT Techniques

The inter-study and inter-center variability in mobilization, graft processing, conditioning, and supportive care provides an opportunity to compare these variables in order to determine the optimal mobilization/graft composition/conditioning/supportive care. International effort combining large datasets will be needed to achieve this goal. Moreover, new techniques should be explored, as exemplified by the fludarabine-based conditioning of Burt et al. [28] or the thiotepa-based conditioning of Henes et al. [89].

### 9.4. Role of Maintenance Therapy

It is unclear whether incidence of relapse/progression after HCT can be lowered by maintenance use of conventional immunomodulatory therapy. We eagerly await the results of a multi-institutional, single-arm trial of maintenance with MMF until 2 years post-HCT (STAT, NCT01413100) [157].

### 9.5. Cellular Therapies Other Than AutoHCT

Finally, other cellular therapies such as alloHCT [155], chimeric antigen receptor (CAR) T cells, and mesenchymal stromal cells [158] need to be explored.

For alloHCT, the use of low-intensity conditioning, resulting in non-fatality of graft rejection, with ATG plus post-transplant cyclophosphamide-based GVHD prophylaxis, resulting in a low incidence of acute and chronic GVHD, is theoretically attractive. It has been successfully used for hemoglobinopathies and aplastic anemia [147,148,149,150,151].

For CAR T cells, as rituximab appears effective for SSc treatment, it may be worth exploring B cell-depleting CAR T cells, e.g., CD19-directed CAR T cells that are commercially available for lymphomas and acute lymphoblastic leukemia. To this end, we eagerly await the results of a clinical trial which studies the effect of CD19/BCMA-directed CAR T cells in the treatment of systemic sclerosis (NCT05085444) [159]. 

## 10. Summary

In summary, it is encouraging that after decades of only minor progress in SSc therapy, there is a treatment associated with improved survival and QOL. Nevertheless, much work needs to be conducted to further reduce HCT-related morbidity and mortality and to reduce the incidence of post-HCT relapse and sAD. This could come from basic/translational science elucidating the mechanism of the beneficial effect of HCT or from new clinical studies. Pending further progress, we wish to emphasize the need for the early diagnosis and early referral of dcSSc patients for consideration of HCT, to avoid irreversible or only minimally reversible organ damage. The ideal candidate for HCT appears to be a patient with an early but rapidly progressing disease with only mild internal organ involvement.

## Figures and Tables

**Figure 1 cells-11-03912-f001:**
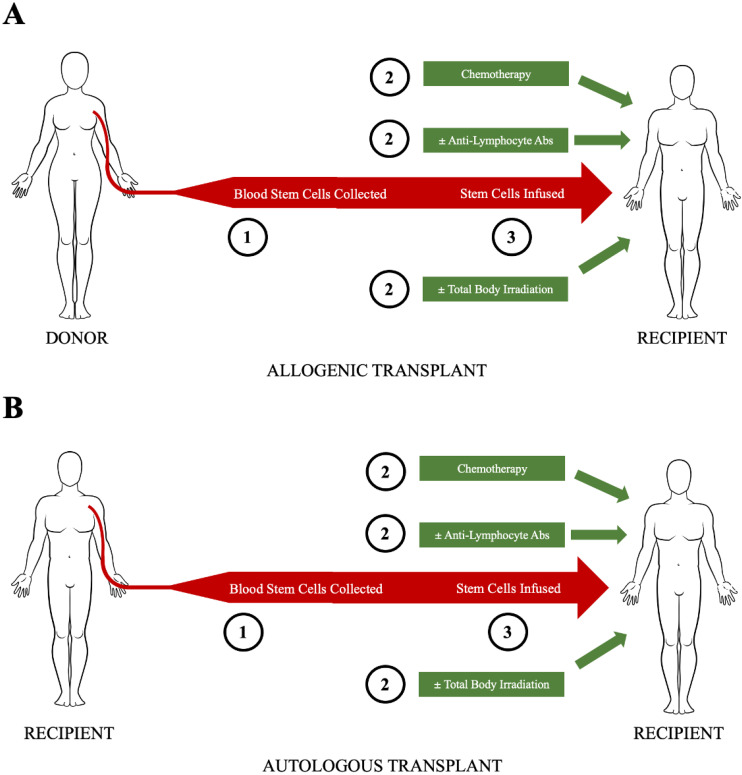
Schematic of hematopoietic stem cell transplantation ((**A**), allogeneic, (**B**), autologous). Step No. 1 is stem cell mobilization (from marrow to blood) and collection by apheresis. Step No. 2 is chemotherapy, usually with anti-lymphocyte antibodies (e.g., antithymocyte globulin) and sometimes with total body irradiation. Step No. 3 is the infusion of the stem cells, usually fresh stem cells in the allogeneic setting and usually thawed, previously cryopreserved stem cells in the autologous setting. Step No. 4 (not shown here) is post-transplant supportive care (e.g., transfusions for low cell counts, antibiotics for infections).

**Figure 2 cells-11-03912-f002:**
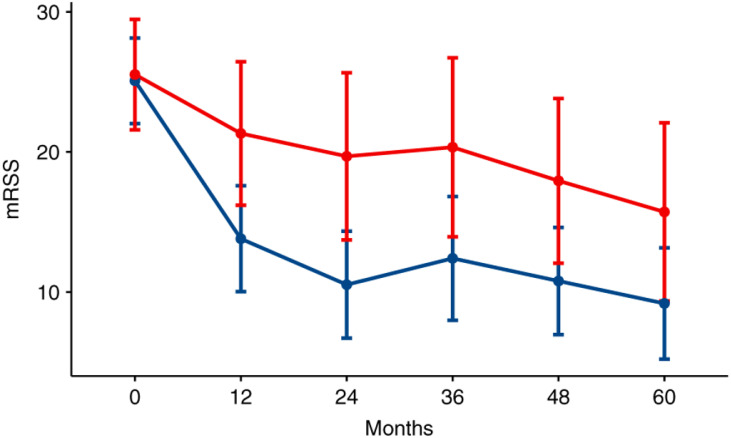
Change in modified Rodnan skin score [mRSS], where larger scores indicate worse skin tightness) over 60 months from randomization in 26 patients treated with autologous hematopoietic cell transplantation (auto-HCT: blue line) vs. 23 patients treated with conventional therapy (12 months of IV cyclophosphamide: red line) [67]. The skin tightness improved faster and to a greater degree after HCT compared to the conventional therapy. Reproduced with permission from Springer Nature, Bone Marrow Transplantation; published by Springer Nature, 2021.

**Figure 3 cells-11-03912-f003:**
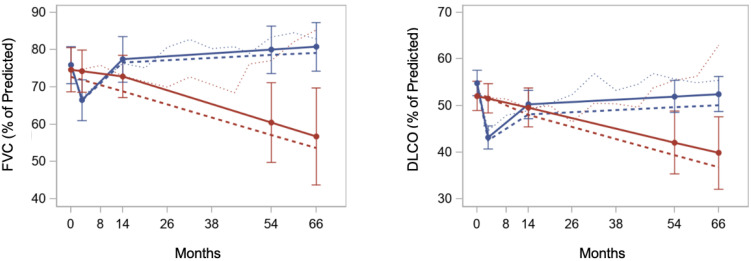
Longitudinal trends over 66 months from randomization for Forced Vital Capacity (FVC: left) and diffusing lung capacity for carbon monoxide (DLCO: right) in patients undergoing autologous hematopoietic cell transplantation (autoHCT: blue line) vs. conventional therapy (12 monthly infusions of cyclophosphamide: red line) [68]. AutoHCT patients had slightly improved FVC at 54 and 66 months compared to baseline, whereas the FVC worsened in the conventionally treated patients. Similar trends were observed for DLCO. Reproduced with permission from Wiley Global Publishing, *Arthritis Care and Research*, published by Wiley Publishing Global, 2021.

**Figure 4 cells-11-03912-f004:**
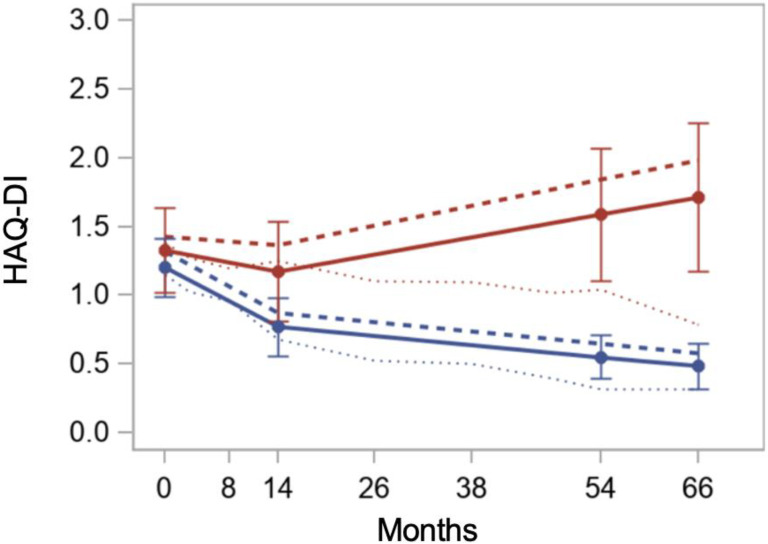
Longitudinal trend over 66 months from randomization for quality of life, as measured by the Health Assessment Questionnaire Disability Index (HAQ-DI) for patients undergoing autologous hematopoietic cell transplantation (autoHCT: blue line) vs. conventional therapy (12 monthly infusions of cyclophosphamide: red line) [68]. The HAQ-DI improved (became lower) in the autoHCT patients, whereas it worsened in the conventionally treated patients. Reproduced with permission from Wiley Global Publishing, *Arthritis Care and Research*, published by Wiley Publishing Global, 2021.

**Figure 5 cells-11-03912-f005:**
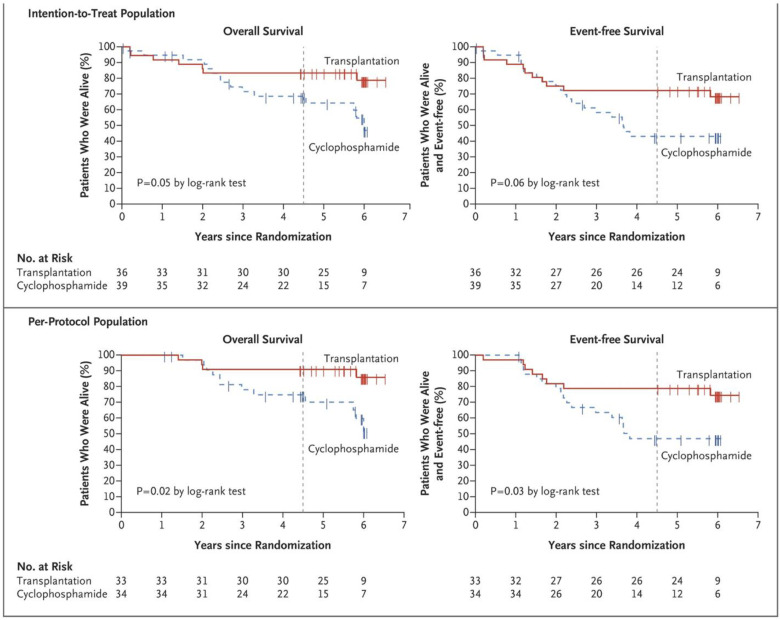
Survival data from the SCOT trial comparing autologous hematopoietic cell transplantation (autoHCT: red line) to conventional therapy (12 monthly doses of IV cyclophosphamide: blue line) [23]. The graphs show the overall survival and the event (organ failure)-free survival in the intention-to-treat analysis (including all participants who had undergone randomization: *top*) and per-protocol analysis (including only participants who received HCT or completed ≥9 doses of cyclophosphamide: bottom). The vertical dashed line denotes the 54-month time point. Both overall survival and event-free survival were significantly higher in autoHCT patients in the per-protocol analysis and trended to be higher in the intention-to-treat analysis. Reproduced with permission from Massachusetts Medical Society, *The New Engl. J. Med.,* published by Massachusetts Medical Society, 2018.

**Table 1 cells-11-03912-t001:** Effects of HCT on multiple manifestations/aspects of SSc *.

Manifestation/Aspect of SSc		Effect of HCT	Level of Evidence **	
Mortality		Improvement	High [21,22,23]	
Quality of life		Improvement	High [21,22,23,65,71,72,73]	
Skin tightness (mRSS) [74]		Improvement	High [21,22,23,24,25,44,65,66,72,75,76,77,78,79]	
Pain [80,81,82]		Improvement	Medium [71]	
Lung function (FVC) [83,84]		Mild improvement	High [21,22,23,28,75,78,85,86,87]	
Interstitial lung disease (CT) [88]		Improvement of inflammation, probably not fibrosis	Medium [28,85,86,87,89,90,91]	
Pulmonary arterial hypertension		Preventing PAH; effect on established PAH is unknown	High [23]	
Heart failure [92,93,94]		Preventing or delaying HF; effect on established HF is unknown	High [23,95]	
Esophageal volume/dilation (CT)		Worsening	Medium [96]	
Esophageal motility		Stabilization or mild improvement(?) if pre-transplant hypomotility; no improvement if pre-transplant amotility	Medium, but unpublished [M.W. et al., publication in progress]	
Gastrointestinal symptoms [97]	Improvement	Medium [71]		
Renal function [98]	Worsening	High, but shown in only 1 of 3 randomized studies [22]		
Range of motion of joints		Improvement	Medium [77]	
Hand grip strength	Improvement	Medium [77]		
Exercise capacity	Improvement	Medium [99]		
Myositis	May improve	Low [100]		
Peripheral neuropathy	May improve	Low [101]		
Nailfold capillaries [102]	May improve	Medium [103]		
Capillary density on skin biopsy			No improvement	Medium [104]
Raynaud’s phenomenon	May improve late after HCT	Medium [71]		
Calcinosis	May improve	Low [105]		

* This is a summary table. For details, see Clinical Results section and Appendix A online. ** Levels of evidence are defined as High (based on at least one prospective randomized study), Moderate (based on at least one retrospective case series study), and Low (based on only case reports or anecdotal observations). Abbreviations: HCT = hematopoietic cell transplantation, SSc = systemic sclerosis, mRSS = modified Rodnan skin score, FVC = forced vital capacity, CT = computer tomography, HF = heart failure.

**Table 2 cells-11-03912-t002:** Selection criteria for HCT, including conventionally accepted criteria for transplant, and studies exploring expanded criteria.

	Conventional Criteria	Expanded Criteria
Age	18–65 Years of Age	No Expansion
Disease Duration	≤5 years from first non-Raynaud’s manifestation	Shah et al. 2021 [109]: - Included one patient with 15 y disease duration Burt et al. 2021 [28]: - Included patients with up to 17 y disease duration
SSc Subtype/mRSS	dcSSc - with mRSS ≥ 20 with or without visceral organ involvement (other than esophagus), particularly if mRSS is worsening or accompanied by inflammation (elevated CRP or ESR) - If mRSS < 20, then concomitant interstitial lung disease is required with DLCO or FVC < 80% or worsening (e.g., FVC decline by 10% over 12–18 months) Mild involvement of the heart or the kidneys may be an acceptable indication if no interstitial lung disease	Burt et al. 2021: - Included 10 patients with lcSSc (mRSS as low as 3) who had concomitant pulmonary involvement Farge et al. 2021 [13]: - Included lcSSC among suggested criteria if concomitant severe/progressive interstitial lung disease
Smoking	Non-current smokers	No expansion
Pregnancy	Non-pregnant individuals	No expansion
Cardiopulmonary Involvement	- PaO_2_ > 60 mmHg at rest without supplemental oxygen - PaCO_2_ < 50 mmHg - PAPsyst < 40 mmHg or PAPmean < 25 mmHg at baseline - PAPsyst < 45 mmHg or PAPmean < 30 mmHg after fluid challenge - DLCO or FVC > 40% predicted - LVEF > 45% - No refractory CHF - No evidence of severe non-revascularized coronary artery disease - No uncontrolled ventricular arrhythmia - No large pericardial effusion or tamponade	Shah et al. 2021: - Included 2 patients with DLCO 29% and 34% predicted Burt et al. 2021: - Designed a study using fludarabine-based conditioning for patients who would not meet conventional selection criteria, e.g., did not exclude patients with PASP > 40 mmHg at rest or >45 mmHg with fluid challenge, mPAP > 25 mmHg at rest or > 30 mmHg with fluid challenge, or diastolic interventricular septal flattening or septal bounce; cardiac tamponade was an exclusion criterion, and the study accrued 42 patients, including: - 13 oxygen-dependent patients - 14 patients with mPAP > 30 mmHg with fluid challenge - 14 patients with ventricular arrhythmias - 5 patients with a conduction block - 9 patients with elevated basic natriuretic peptide - 10 patients with late gadolinium enhancement on cardiac MRI - 15 patients with pericardial effusion - FVC was as low as 32%; DLCO as low as 24%
Hepatic Involvement	No evidence of persistent or progressive hepatic impairment defined as persistent increase in rate or twice-normal transaminases/bilirubin	No expansion
Renal Involvement	GFR > 40 mL/min	No expansion
Hematolymphatic System Status	No evidence of persistent neutropenia (neutrophils <0.5 × 10^9^/L), thrombocytopenia (<50 × 10^9^/L), or CD4 lymphopenia (<200/mm^3^)	No expansion
Nutritional Status	BMI ≥ 18 kg/m^2^, Albumin ≥ 20 mg/L	No expansion
Other	No active neoplasia or concomitant myelodysplasia, no acute or chronic uncontrolled infection, good patient compliance	No expansion

Abbreviations: Diffuse SSc (dcSSc), limited SSc (lcSSC), mRSS (modified Rodnan score), diffusing capacity for carbon monoxide (DLCO), forced vital capacity (FVC), chest X-ray (CXR), high-resolution CT scan (HRCT), erythrocyte sedimentation rate (ESR), hemoglobin (HgB), hypertension (HTN), congestive heart failure (CHF), PaO_2_ (partial pressure of oxygen), PaCO_2_ (partial pressure of carbon dioxide), PAP (pulmonary arterial pressure), PAPm (mean pulmonary arterial pressure), glomerular filtration rate (GFR).

## Data Availability

No new data reported.

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
