# Peer review of "Hematopoietic Cell Transplantation for Systemic Sclerosis—A Review"

_cells, 2022, doi:10.3390/cells11233912_

Round 1

Reviewer 1 Report

The article ”Hematopoietic Cell Transplantation for Systemic Sclerosis – A  Review” is a difficult-to-read one but with a high scientific level dedicated to experts in the same restricted area of interest. In this sense, at least in the introduction, it is necessary to make it more accessible for a larger spectrum of readers, even those not directly involved in HCT.

My first concern was if the article fits in the Cells Journal topic related to cell biology and physiology, molecular biology, and biophysics, with a major focus on experimental cytology rather than clinical and epidemiological studies. By Proposed Mechanisms of Action of autoHCT on SSc - this concern was partially excluded, but it seems necessary to highlight these aspects as the central part of the article in order to be aligned with the journal's aim.

Future Works include some Discussion and this aspect influences the high level of expertise that requires a qualified reader. 

The article is too long and needs to be synthesized regarding some parts. Maybe tables can be part of the supplementary data.

Will be advisable to include some figures in order to improve the quality of the article and to underline the important outcomes of the article.

Reviewer 2 Report

overall - nicely done and complete review of the topic

comments;

line 29 - should it read ...'outcomes are worse in men" (add the word "men")

line 50 -should it read..."replace an autoreactive immune system with a healthy one." (add the word "one")

Note: the modified Rodnan skin score (mRSS) is a skin thickness score - it is not, and has never been, a skin tightness score. Please replace this as needed throughout the paper.

The tables and figures are ezcellent.

Round 2

Reviewer 1 Report

The manuscript “Hematopoietic Cell Transplantation for Systemic Sclerosis – A Review” was improved through: 

o   adding a paragraph to the Introduction that briefly introduces HCT, as well as a figure introducing HCT.

o   minor changes to improve the readability.

o   “Proposed Mechanisms of Action of autoHCT on SSc” section was made more prominent by moving this section close to the beginning – right after the Introduction.

o   A large section on the effect of HCT on multiple manifestations/ aspects of SSc has been moved into Supplementary Material online. It is now also summarized in a condensed way in a new table (Table 1).

o   Two tables have been moved to Supplementary Material online

o   A new figure (Figure 1) has been added to four existing figures

After the above-presented changes and rereading the manuscript, I consider that the revised version is properly improved and all my concerns and comments were adequately addressed, so I do not have further comments and conclude for acceptance for publication in the present form.